# Cooperative Learning of Audio and Video Models from Self-Supervised Synchronization

**Bruno Korbar**
Dartmouth College
bruno.18@dartmouth.edu

**Du Tran**
Facebook Research
trandu@fb.com

**Lorenzo Torresani**
Dartmouth College
LT@dartmouth.edu

## Abstract

There is a natural correlation between the visual and auditive elements of a video. In this work we leverage this connection to learn general and effective models for both audio and video analysis from self-supervised temporal synchronization. We demonstrate that a calibrated curriculum learning scheme, a careful choice of negative examples, and the use of a contrastive loss are critical ingredients to obtain powerful multi-sensory representations from models optimized to discern temporal synchronization of audio-video pairs. Without further finetuning, the resulting audio features achieve performance superior or comparable to the state-of-the-art on established audio classification benchmarks (DCASE2014 and ESC-50). At the same time, our visual subnet provides a very effective initialization to improve the accuracy of video-based action recognition models: compared to learning from scratch, our self-supervised pretraining yields a remarkable gain of +19.9% in action recognition accuracy on UCF101 and a boost of +17.7% on HMDB51.

## 1 Introduction

Image recognition has undergone dramatic progress since the breakthrough of AlexNet [1] and the widespread availability of progressively large datasets such as Imagenet [2]. Models pretrained on Imagenet [2] have enabled the development of feature extractors achieving strong performance on a variety of related still-image analysis tasks, including object detection [3, 4], pose estimation [5, 6] and semantic segmentation [7, 8]. Deep learning approaches in video understanding have been less successful, as evidenced by the fact that deep spatiotemporal models trained on videos [9, 10] still barely outperform the best hand-crafted features [11].

Researchers have devoted significant and laudable efforts in creating video benchmarks of much larger size compared to the past [9, 12, 13, 14, 15, 16], both in terms of number of examples as well as number of action classes. The growth in scale has enabled more effective end-to-end training of deep models and the finer-grained definition of classes has made possible the learning of more discriminative features. This has inspired a new generation of deep video models [17, 18, 19] greatly advancing the field. But such progress has come at a high cost in terms of time-consuming manual annotations. In addition, one may argue that future significant improvements by mere dataset growth will require scaling up existing benchmarks by one or more orders of magnitude, which may not be possible in the short term.

In this paper, we explore a different avenue by introducing a self-supervision scheme that does not require any manual labeling of videos and thus can be applied to create arbitrarily-large training sets for video modeling. Our idea is to leverage the natural synergy between the audio and the visual channels of a video by introducing a self-supervised task that entails deciding whether a given audio sample and a visual sequence are either "in-sync" or "out-of-sync." This is formally defined as a binary classification problem which we name "Audio-Visual Temporal Synchronization" (AVTS). We propose to address this task via a two-stream network, where one stream receives audio as input and

the other stream operates on video. The two streams are fused in the late layers of the network. This design induces a form of *cooperative learning* where the two streams must learn to "work together" in order to improve performance on the synchronization task.

We are not the first to propose to leverage correlation between video and audio as a self-supervised mechanism for feature learning [20, 21, 22]. However, unlike prior approaches that were trained to learn semantic correspondence between the audio and a single frame of the video [21] or that used 2D CNNs to model the visual stream [22], we propose to feed video clips to a 3D CNN [19] in order to learn spatiotemporal features that can model the correlation of sound and *motion* in addition to appearance within the video. We note that AVTS differs conceptually from the "Audio-Visual Correspondence" (AVC) proposed by Arandjelovic and Zisserman [21, 22]. In AVC, negative training pairs were formed by drawing the audio and the visual samples from distinct videos. This makes it possible to solve AVC purely based on semantic information (e.g., if the image contains a piano but the audio includes sound from a waterfall, the pair is obviously a negative sample). Conversely, in our work we train on negative samples that are "hard," i.e., represent out-of-sync audio and visual segments sampled from the same video. This forces the net to learn relevant temporal-sensitive features for both the audio and the video stream in order to recognize *synchronization* as opposed to only semantic correspondence.

Temporal synchronization is a harder problem to solve than semantic correspondence, since it requires to determine whether the audio and the visual samples are not only semantically coherent but also temporally aligned. To ease the learning, we demonstrate that it is beneficial to adopt a *curriculum learning* strategy [23], where harder negatives are introduced after an initial stage of learning on easier negatives. We demonstrate that using curriculum learning further improves the quality of our features for all the downstream tasks considered in our experiments.

The audio and the visual components of a video are processed by two distinct streams in our network. After learning, it is then possible to use the two individual streams as feature extractors or models for each of the two modalities. In our experiments we study several such applications, including pretraining for action recognition in video, feature extraction for audio classification, as well as multisensory (visual and audio) video categorization. Specifically, we demonstrate that, without further finetuning, the features computed from the last convolutional layer of the audio stream yield performance on par with or better than the state-of-the-art on established audio classification benchmarks (DCASE2014 and ESC-50). In addition, we show that our visual subnet provides a very effective initialization to improve the performance of action recognition networks on medium-size video classification datasets, such as HMDB51 [24] and UCF101 [25]. Furthermore, additional boosts in video classification performance can be obtained by finetuning multisensory (audio-visual) models from our pretrained two-stream network.

## 2   Technical Approach

In this section we provide an overview of our approach for Audio-Visual Temporal Synchronization (AVTS). We begin with a formal definition of the problem statement. We then introduce the key-features of our model by discussing their individual quantitative contribution towards both AVTS performance and accuracy on our downstream tasks (action recognition and audio classification).

### 2.1   Audio-Visual Temporal Synchronization (AVTS)

We assume we are given a training dataset $\mathcal{D} = \{(a^{(1)}, v^{(1)}, y^{(1)}), \ldots, (a^{(N)}, v^{(N)}, y^{(N)})\}$ consisting of $N$ labeled audio-video pairs. Here $a^{(n)}$ and $v^{(n)}$ denote the audio sample and the visual clip (a sequence of RGB frames) in the $n$-th example, respectively. The label $y^{(n)} \in \{0, 1\}$ indicates whether the audio and the visual inputs are "in sync," i.e., if they were sampled from the same temporal slice of a video. If $y^{(n)} = 0$, then $a^{(n)}$ and $v^{(n)}$ were taken either from different temporal segments of the same video, or possibly from two different videos, as further discussed below. The audio input $a^{(n)}$ and the visual clip $v^{(n)}$ are sampled to span the same temporal duration.

At a very high level, the objective of AVTS is to learn a classification function $g(.)$ that minimizes the empirical error, i.e., such that $g(a^{(n)}, v^{(n)}) = y^{(n)}$ on as many examples as possible. However, as our primary goal is to use AVTS as a self-supervised proxy for audio-visual feature learning, we define $g(.)$ in terms of a two-stream network where the audio and the video input are separately processed by

an audio subnetwork $f_a(.)$ and a visual subnetwork $f_v(.)$, providing a feature representation for each modality. The function $g(f_a(a^{(n)}), f_v(v^{(n)}))$ is then responsible to fuse the feature information from both modalities to address the synchronization task. An illustration of our two-stream network design is provided in Fig. 2. The technical details about the two streams are provided in subsection 2.5.

## 2.2 Choice of Loss Function

A natural choice is to adopt the cross-entropy loss as learning objective, since this would directly model AVTS as a binary classification problem. However, we found it difficult to achieve convergence under this loss when learning from scratch. Inspired by similar findings in Chung et al [26], we discovered experimentally that more consistent and robust optimization can be obtained by minimizing the contrastive loss, which was originally introduced for training Siamese networks [27] on same-modality input pairs. In our setting, we optimize the audio and video streams to produce small distance on positive pairs and larger distance on negative pairs, as in [26]:

$$E = \frac{1}{N} \sum_{n=1}^{N} (y^{(n)}) ||f_v(v^{(n)}) - f_a(a^{(n)})||_2^2 + (1 - y^{(n)}) \max(\eta - ||f_v(v^{(n)}) - f_a(a^{(n)})||_2, 0)^2 \quad (1)$$

where $\eta$ is a margin hyper-parameter. Upon convergence, AVTS prediction on new test examples $(a, v)$ can be addressed by simply thresholding the distance function, i.e., by defining $g(f_a(a), f_v(v)) \equiv 1\{||f_v(v^{(n)}) - f_a(a^{(n)})||_2 < \tau\}$ where $1\{.\}$ denotes the logical indicator function and $\tau$ is a set threshold. We also tried adding one or more fully connected (FC) layers on top of the learned feature extractors and fine-tuning the entire network end-to-end with respect to a cross-entropy loss. We found both these approaches to perform similarly on the AVTS task, with a slight edge in favor of the fine-tuning solution (see details in subsection 3.2). However, on downstream tasks (action recognition and audio classification), we found AVTS fine-tuning using the cross-entropy loss to yield no further improvement after the contrastive loss optimization.

## 2.3 Selection of Negative Examples

We use an equal proportion of positive and negative examples for training. We generate a *positive* example by extracting the audio and the visual input from a randomly chosen video clip, so that the video frames correspond in time with the audio segment. We consider two main types of *negative* examples. *Easy negatives* are those where the video frames and the sound come from two different videos. *Hard negatives* are those where the pair is taken from the same video, but there is at least half a second time-gap between the audio sample and the visual clip. The purpose of hard negatives is to train the network to recognize temporal synchronization as opposed to mere semantic correspondence between the audio and the visual input. An illustration of a positive example, and the two types of hard negatives is provided in Fig. 1. We have also tried using *super-hard negatives* which we define as examples where the audio and the visual sequence overlap for a certain (fixed) temporal extent.

Not surprisingly, we found that including either hard or super-hard negatives as additional training examples was detrimental when the negative examples in the test set consisted of only "easy" negatives (e.g., the AVTS accuracy of our system drops by about 10% when using a negative training set consisting of 75% easy negatives and 25% hard negatives compared to using negative examples that are all easy). Less intuitively, at first we found that introducing hard or super-hard negatives in the training set degraded also the quality of audio and video features with respect to our downstream tasks of audio classification and action recognition in video. As further discussed in the next subsection, adopting a curriculum learning strategy was critical to successfully leverage the information contained in hard negatives to achieve improved performance in terms of AVTS and downstream tasks.

## 2.4 Curriculum Learning

We trained our system from scratch with easy negatives alone, with hard negatives alone, as well as with fixed proportions of easy and hard negatives. We found that when hard negatives are introduced from the beginning — either fully or as a proportion — the objective is very difficult to optimize and test results on the AVTS task are consequently poor. However, if we introduce the hard negatives after the initial optimization with easy negatives only (in our case between 40th and 50th epoch), fine-tuning using some harder negatives yields better results in terms of both AVTS accuracy as well

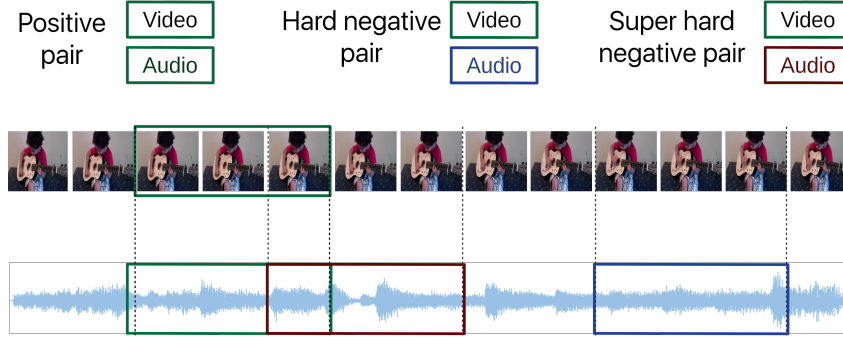

Figure 1: Illustration of a positive example, a "hard" negative and "super-hard" negative. "Easy" negative are not shown here: they involve taking audio samples and visual clips from different videos. Easy negatives can be recognized merely based on semantic information, since two distinct videos are likely to contain different scenes and objects. Our approach uses hard negatives (audio and visual samples taken from different slices of the same video) to force the network to recognized synchronization, as opposed to mere semantic correspondence.

as performance on our downstream tasks (audio classification and action recognition). Empirically, we obtained the best results when fine-tuning with a negative set consisting of 25% hard negatives and 75% easy negatives. For a preview of results see Table 1, which outlines the difference in AVTS accuracy when training using curriculum learning as opposed to single-stage learning. Even more remarkable are the performance improvements enabled by curriculum feature learning on the downstream tasks of audio classification and action recognition (see Table 4).

## 2.5 Architecture Design

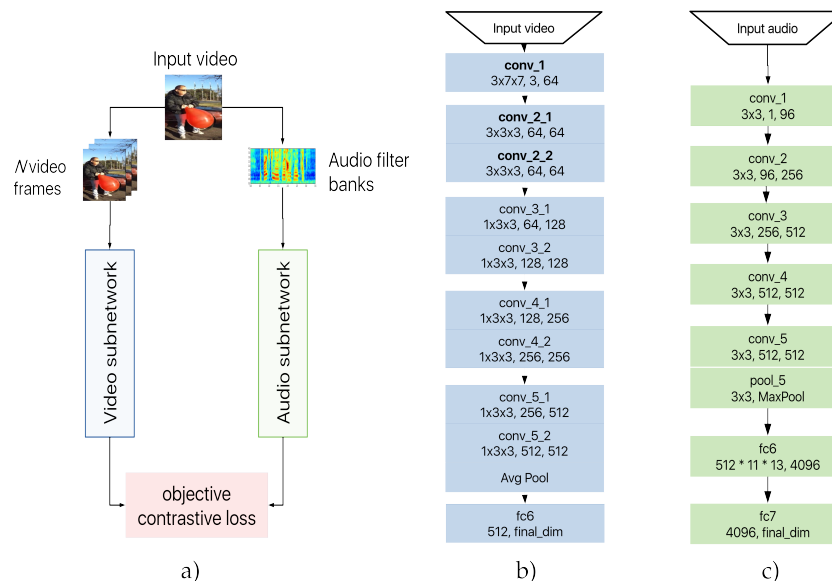

Figure 2: Our architecture design. The complete model for AVTS training can be viewed in (a). The video subnetwork (shown in (b)) is a MC$x$ network [19] using 3D convolutions in the early layers, and 2D convolutions in the subsequent layers. The audio subnetwork (shown in (c)) is the VGG model used by Chung and Zisserman [26].

As illustrated in Fig. 2(a), our network architecture is composed of two main parts: the audio subnetwork and the video subnetwork, each taking its respective input. Our video subnetwork (shown in Fig. 2(b)) is based on the mixed-convolution (MC$x$) family of architectures [19]. A MC$x$

Table 1: AVTS accuracy achieved by our system on the Kinetics test set, which includes negatives of only "easy" type, as in [21]. The table shows that curriculum learning with a mix of easy and hard negatives in a second stage of training leads to a significant gain in accuracy.

| Method | Negative type | Epochs | Accuracy (%) |
|---|---|---|---|
| **Single learning stage** | easy | 1 - 90 | 69.0 |
| | 75% easy, 25% hard | 1 - 90 | 58.9 |
| | hard | 1 - 90 | 52.3 |
| | easy | 1 - 50 | 67.2 |
| **Curriculum learning** (i.e., second learning stage applied after a first stage of 1-50 epochs with easy negatives only) | 75% easy, 25% hard | 51 - 90 | **78.4** |
| | hard | 51 - 90 | 65.7 |

network uses a sequence of $x$ 3D (spatiotemporal) convolutions in the early layers, followed by 2D convolutions in the subsequent layers. The intuition is that temporal modeling by 3D convolutions is particularly useful in the early layers, while the late layers responsible for the final prediction do not require temporal processing. MC$x$ models were shown to provide a good trade off in terms of video classification accuracy, number of learning parameters, and runtime efficiency (see [19] for further details). We found that within our system MC3 yields the best performance overall and is used whenever not specified otherwise. Note that our architecture is a simplified version of the original network discussed in [19], as it lacks residual connections and differs in terms of dimensionality in the final FC layer. The input in our video subnetwork are video clips of size $(3 \times t \times h \times w)$, where 3 refers to the RGB channels of each frame, $t$ is the number of frames, and $h, w$ are the height and width, respectively. For the audio stream, we use the processing and the architecture described by Chung and Zisserman [26]: the audio is first converted to the MP3 format and FFT filterbank features are computed and passed through a VGG-like convolutional architecture. The specifications of the audio subnetwork are provided in Figure 2(c). Further implementation details can be found in [26].

## 3 Experiments

### 3.1 Implementation Details

**Input preprocessing.** The starting frame of each clip is chosen at random within a video. The length of each clip is set to $t = 25$ frames. This results in a clip duration of 1 second on all the datasets considered here except for HMDB51, which uses a frame rate different from 25 fps (clip duration on HMDB51 is roughly 1.2 seconds). Standard spatial transformations (multi-scale random crop, random horizontal flip, and $\mathbf{Z}$ normalization) are applied to all frames of the clip at training time. FFT filterbank features are extracted from the audio sample, and $\mathbf{Z}$ normalization is applied. The FFT filterbank parameters are set as follows: window length to $0.02$, window step to $0.01$, FFT size to $1024$, and number of filters to $40$. Mean and standard deviation for normalization are extracted over a random $20\%$ subset of the training dataset.

**Training details.** Hyper-parameter $\eta$ in Eq. 1 is set to 0.99. We train the complete AVTS network end-to-end using stochastic gradient descent with initial learning rate determined via grid search. Training is done on a four-GPU machine with a mini-batch of 16 examples per GPU. The learning rate is scaled by 0.1 each time the loss value fails to decrease for more than 5 epochs.

### 3.2 Evaluation on Audio-Visual Temporal Synchronization (AVTS)

We first evaluate our approach on the AVTS task. We experimented with training our model on several datasets: Kinetics [12], SoundNet [20], and AudioSet [28]. We tried different ways to combine the information from the two streams after training from scratch the network shown in Fig. 2 with contrastive loss. The best results were obtained by concatenating the outputs of the two subnetworks, and by adding two fully connected layers (containing 512 and 2 units, respectively). The resulting network was then finetuned end-to-end for the binary classification AVTS task using cross-entropy loss. In order to have results completely comparable to those of Arandjelovic *et al.* [21], we use their same test set including only negatives of "easy" type.

Table 2: Action recognition accuracy (%) on UCF101 [25] and HMDB51 [24] using AVTS as a self-supervised pretraining mechanism. Even though our pretraining does not leverage any manual labels, it yields a remarkable gain in accuracy compared to learning from scratch (+19.9% on UCF101 and +17.7% on HMDB51, for MC3). As expected, making use of Kinetics action labels for pretraining yields further boost. But the accuracy gaps are not too large (only +1.5% on UCF101 with MC3) and may potentially be bridged by making use of a larger pretraining dataset, since no manual cost is involved for our procedure. Additionally, we show that our method generalizes to different families of models, such as I3D-RGB [18]. Rows marked with * report numbers as listed in the I3D-RGB paper, which may have used a slightly different setup in terms of data preprocessing and evaluation.

| Video Network Architecture | Pretraining Dataset | Pretraining Supervision | UCF101 | HMDB51 |
|---|---|---|---|---|
| MC2 | none | N/A | 67.2 | 41.2 |
| MC2 | Kinetics | self-supervised (AVTS) | 83.6 | 54.3 |
| MC2 | Kinetics | fully supervised (action labels) | 87.9 | 62.0 |
| MC3 | none | N/A | 69.1 | 43.9 |
| MC3 | Kinetics | self-supervised (AVTS) | 85.8 | 56.9 |
| MC3 | Audioset | self-supervised (AVTS) | 89.0 | 61.6 |
| MC3 | Kinetics | fully supervised (action labels) | 90.5 | 66.8 |
| I3D-RGB | none | N/A | 57.1 | 40.0 |
| I3D-RGB | Kinetics | self-supervised (AVTS) | 83.7 | 53.0 |
| I3D-RGB* | Imagenet | fully supervised (object labels) | 84.5 | 49.8 |
| I3D-RGB* | Kinetics | fully supervised (action labels) | 95.1 | 74.3 |
| I3D-RGB* | Kinetics + Imagenet | fully supervised (object+action labels) | 95.6 | 74.8 |

Table 1 summarizes our AVTS results when training on the Kinetics training set and testing on the Kinetics test set. We can clearly see that inclusion of hard negatives in the first stage of training is deleterious. However, curriculum learning with a 75/25% mix of easy/hard negatives in the second stage (after training only on easy negatives in the first stage) yields a remarkable gain of 9.4% in AVTS accuracy. However, we found detrimental to use super-hard negatives in any stage of training, as these tend to make the optimization overly difficult.

As noted in Section 2.1, performing AVTS classification by directly thresholding the contrastive loss distance (i.e., using classifier $1\{||f_v(v) - f_a(a)||_2 < \tau\}$) produces comparable performance when the test set includes negatives of only easy type (76.1% on Kinetics). However, we found it to be largely inferior when evaluated on a test set including a mix of 25/75% of hard and easy negatives (65.6% as opposed to 70.3% when using the net finetuned with cross-entropy).

In terms of comparison with the $L^3$-Net of Arandjelovic and Zisserman [21], our approach does consistently better: 78% vs 74% on Kinetics and 86% vs 81% on Audioset [28] (learning on the training split of Audioset and testing on the test split of Audioset). Inclusion of hard negatives during training allows us to maintain high performance even when hard negatives are included in the testing set (70% on Kinetics), whereas we found that the performance of $L^3$-Net drops drastically when hard-negatives are included in the test set (57% for $L^3$-Net vs 70% for our AVTS).

We stress, however, that performance on the AVTS task is **not** our ultimate objective. AVTS is only a proxy for learning rich audio-visual representations. In the following sections, we present evaluations of our AVTS audio and visual subnets on downstream tasks.

### 3.3 Evaluation of AVTS as a Pretraining Mechanism for Action Recognition

In this section we assess the ability of the AVTS procedure to serve as an effective pretraining mechanism for video-based action recognition models. For this purpose, after AVTS training with contrastive loss on Kinetics, we fine-tune our video subnetwork on two medium-size action recognition benchmarks: UCF101 [25] and HMDB51 [24]. We note that since AVTS does not require any labels, we could have used any video dataset for pretraining. Here we use Kinetics for AVTS learning, as it will allow us to compare the results of our self-supervised pretraining (i.e., no manual labels) with those obtained by fully-supervised pretraining obtained by making use of action class labels, which are available on Kinetics. We also include results by training the MC-$x$

network from scratch on UCF101 [25] and HMDB51. Finally, we report action recognition results using the I3D-RGB [18] network trained in several ways: learned from scratch, pretrained using our self-supervised AVTS, or pretrained with category labels (using object labels from ImageNet with 2D-to-3D filter inflation [18], using action labels from Kinetics, or using both ImageNet and Kinetics). Results are computed as averages over the 3 train/test splits provided with these two benchmarks.

The results are provided in Table 2. We note that AVTS pretraining provides a remarkable boost in accuracy on both datasets. For the MC3 model, the gain produced by AVTS-pretraining on Kinetics is 16.7% on UCF101 and 13.0% on HMDB51, compared to learning from scratch. This renders our approach practically effective as a pretraining mechanism for video-based classification model, since we stress that **zero** manual annotations were used to produce this gain in accuracy. As expected, pretraining on the Kinetics dataset using action-class labels leads to even higher accuracy on UCF101 and HDMB51. But this leverages the enormous cost of manual labeling over the 500K video clips. Conversely, since our pretraining is self-supervised, it can be applied to even larger datasets at no additional manual cost. Here we investigate the effect of larger self-supervised training sets by performing AVTS pretraining on AudioSet [28], which is nearly 8x bigger than Kinetics. As shown in Table 2, in this case the accuracy of MC3 further improves, reaching 89.0% on UCF101. This is only 1.5% lower than the accuracy obtained by pretraining with full supervision on Kinetics. Conversely, AVTS pretraining on a subset of Audioset having the same size as Kinetics leads to an accuracy of 86.4% on UCF101. This suggests that AVTS pretraining on even larger datasets is likely to lead to further accuracy improvements and may help bridging the remaining gap with respect to methods based on fully-supervised pretraining.

## 3.4 Evaluation of AVTS Audio Features

In this section we evaluate the audio features learned by our AVTS procedure by minimization of the contrastive loss. For this purpose, we take the activations from the `conv_5` layer of the audio subnetwork and test their quality as audio representation on two established sound classification datasets: ESC-50 [29] and DCASE2014 [30]. We extract 10 equally-spaced 2-second-long sub-clips from each full audio sample of ESC-50. For DCASE2014, we extract 60 sub-clips from each full sample since audio samples in this dataset are longer than those of ESC-50. We stress that no finetuning of the audio subnetwork is done for these experiments. We directly train a multiclass one-vs-all linear SVM on the `conv_5` AVTS features to classify the audio events. We compute the classification score for each audio sample by averaging the sub-clip scores in the sample, and then predict the class having higher score.

Table 3 summarizes the results of our approach as well as many other methods on these two benchmarks. We can observe that audio features learned on AVTS generalize extremely well on both of these two sound classification tasks, yielding performance superior or close to the state-of-the-art. From the results in the Table it can be noticed that our audio subnet directly trained from scratch on these two benchmarks performs quite poorly. This indicates that the effectiveness of our approach lies in the self-supervised learning procedure rather than in the net architecture. On ESC-50 our approach not only outperforms other recent self-supervised methods ($L^3$Net [21, 22], and SoundNet [20]), but also marginally surpasses human performance.

## 3.5 Multi-Modal Action Recognition

We also evaluate our approach on the task of multi-modal action recognition, i.e., using both the audio and the visual stream in the video to predict actions. As for AVTS classification, we concatenate the audio and the visual features obtained from our two subnetworks and add two fully connected layers (containing, respectively, 512 and $C$ units, where $C$ is number of classes in the dataset), and then fine-tune the entire system for action recognition with cross entropy loss as the training objective. We tried this approach on UCF101 and compared our results to those achieved by the concurrent self-supervised approach of Owens and Efros [32]. Our method achieves higher accuracy than [32] both when both methods rely on the video-stream only (85.8% vs 77.6%) as well as when both methods use multisensory information (audio and video) from both streams (our model achieves 87.0% while the method of Owens and Efros yields 82.1%).

Table 3: Evaluation of audio features learned with AVTS on two audio classification benchmarks: ESC-50 and DCASE2014. "Our audio subnet" denotes our audio subnet directly trained on these benchmarks. The superior performance of our AVTS features suggest the effectiveness of our approach lies in the self-supervised learning procedure rather than in the net architecture.

| Method | Auxiliary dataset | Auxiliary supervision | # auxiliary examples | ESC-50 accuracy (%) | DCASE2014 accuracy (%) |
|---|---|---|---|---|---|
| SVM-MFCC [29] | none | none | none | 39.6 | - |
| Random Forest [29] | none | none | none | 44.3 | - |
| Our audio subnet | none | none | none | 61.6 | 72 |
| SoundNet [20] | SoundNet | self | 2M+ | 74.2 | 88 |
| $L^3$-Net [21] | SoundNet | self | 2M+ | 79.3 | 93 |
| Our AVTS features | Kinetics | self | 230K | 76.7 | 91 |
| Our AVTS features | AudioSet | self | 1.8M | 80.6 | 93 |
| Our AVTS features | SoundNet | self | 2M+ | **82.3** | **94** |
| *Human performance [21]* | *n/a* | *n/a* | *n/a* | *81.3* | - |
| State-of-the-art (RBM)[31] | none | none | none | **86.5** | - |

## 3.6 Impact of curriculum learning on AVTS and downstream tasks

Table 4 presents results across the many tasks considered in this paper. These results highlight the strong benefit of training AVTS with curriculum learning for both the AVTS task, as well as other applications that use our features (audio classification) or finetune our models (action recognition). We also include results achieved by $L^3$-Net (the most similar competitor) across these tasks. For all tasks, both AVTS and $L^3$-Net are pretrained on Kinetics, except for the evaluations on ESC-50 and DCASE2014 where Flickr-Soundnet [20] is used for pretraining.

For a fair comparison, on UCF101 and HMDB51 we fine-tune the $L^3$-Net using all frames from the training set, and evaluate it by using all frames from the test videos. This means that both $L^3$-Net and our network are pre-trained, fine-tuned, and tested on the same amount of data. The results in this table show that AVTS yields consistently higher accuracy across all tasks.

Table 4: Impact of curriculum learning on AVTS and downstream tasks (audio classification and action recognition). Both $L^3$-Net [21] and our AVTS model are pretrained, fine-tuned (when applicable) and tested on the same amount of data. All numbers are accuracy measures (%).

| Method | AVTS-Kinetics | ESC-50 | DCASE | HMDB51 | UCF101 |
|---|---|---|---|---|---|
| Our AVTS - single stage | 69.8 | 70.6 | 89.2 | 46.4 | 77.1 |
| Our AVTS - curriculum | 78.4 | 82.3 | 94.1 | 56.9 | 85.8 |
| $L^3$-Net | 74.3 | 79.3 | 93 | 40.2 | 72.3 |

## 4 Related work

Unsupervised learning has been studied for decades in both computer vision and machine learning. Inspirational work in this area includes deep belief networks [33], stacked autoencoders [34], shift-invariant decoders [35], sparse coding [36], TICA [37], stacked ISAs [38]. Instead of reconstructing the original inputs as typically done in unsupervised learning, self-supervised learning methods try to exploit *free supervision* from images or videos. Wang *et al.* [39] used tracklets of image patches across video frames as self-supervision. Doersch *et al.* [41] exploited the spatial context of image patches to pre-train a deep ConvNet. Fernando *et al.* [42] used temporal context for self-supervised pre-training, while Misra et al. [43] proposed frame-shuffling as a self-supervised task.

Self-supervised learning can also be done across different modalities. Pre-trained visual classifiers (noisy predictions) were used as supervision for pre-training audio models [20] as well as CNN models with depth images as input [44]. Audio signals were also used to pre-train visual models [45]. Recently, Arandjelovic and Zisserman proposed the Audio-Visual Correspondence (AVC) – i.e., predicting if an audio-video pair is in a true correspondence – as a way to jointly learn both auditive

and visual representations [21]. The approach was subsequently further refined for cross-modal retrieval and sound localization [22]. While these approaches used only a single frame of a video and therefore focused on exploiting the *semantics* of the audio-visual correlation, our method uses a video clip as an input. This allows our networks to learn spatiotemporal features. Furthermore, while in AVC negative training pairs are generated by sampling audio and visual slices from two distinct videos, we purposely include out-of-sync audio-visual pairs as negative examples. This transforms the task from audio-video correspondence to one of temporal synchronization. We demonstrate that this forces our model to take into account the temporal information, as well as the appearance, in order to exploit the correlation of sound and motion within the video. Another benefit of the temporal synchronization task is that it allows us to control the level of difficulty in hard negative examples, thus allowing us to develop a curriculum learning strategy which further improves our learned models. Our ablation study shows that both technical contributions (temporal synchronization and curriculum learning) lead to superior audio and video models for classification.

Our approach is also closely similar to the approach described in the work of Chung and Zisserman [26], where the problem of audio-visual synchronization was empirically studied within the application of correlating mouth motion and speech. Here we broaden the scope of the study to encompass arbitrary human activities and experimentally evaluate the task as a self-supervised mechanism for general audio-visual feature learning. Our method is also related to that of Izadinia *et al.* [40] who used canonical correlation analysis to identify moving-sounding objects in video and to solve the problem of audio-video synchronization. Our work is concurrent with that of Zhao et al. [46] and that of Owens and Efros [32]. The former is focused on the task of spatially localizing sound in video as well as the problem of conditioning sound generation with image regions. Conversely, we use audio-visual samples for model learning. The work of Owens and Efros [32] is similar in spirit to our own but we present stronger experimental results on comparable benchmarks (e.g., UCF101) and our technical approach differs substantially in the use of hard-negatives, curriculum learning and the choice of contrastive loss as learning objective. We demonstrated in our ablation study that all these aspects contribute to the strong performance of our system. Finally we note that our two-stream design enables the application of our model to single modality (i.e., audio-only or video-only) after learning, while the network of Owens and Efros requires both modalities as input.

## 5    Conclusions

In this work we have shown that the self-supervised mechanism of audio-visual temporal synchronization (AVTS) can be used to learn general and effective models for both the audio and the vision domain. Our procedure performs a form of cooperative training where the "audio stream" and the "visual stream" learn to work together towards the objective of synchronization classification. By training on videos (as opposed to still-images) and by including out-of-sync audio-video pairs as "hard" negatives we force our model to address the problem of audio-visual synchronization (i.e., are audio and video temporally aligned?) as opposed to mere semantic correspondence (i.e., are audio and video recorded in the same semantic setting?). We demonstrate that this leads to superior performance of audio and visual features for several downstream tasks. We have also shown that curriculum learning significantly improves the quality of the features on all end tasks.

While in this work we trained AVTS on established, labeled video dataset in order to have a direct comparison with fully-supervised pretraining methods, our approach is self-supervised and does not require any manual labeling. This opens up the possibility of self-supervised pretraining on video collections that are much larger than any existing labeled video dataset and that may be derived from many different sources (YouTube, Flickr videos, Facebook posts, TV news, movies, etc.). We believe that this may yield further improvements in the generality and effectiveness of our models for downstream tasks in the audio and video domain and it may help bridge the remaining gap with respect to fully-supervised pretraining that rely on costly manual annotations.

## Acknowledgements

This work was funded in part by NSF award CNS-120552. We gratefully acknowledge NVIDIA and Facebook for the donation of GPUs used for portions of this work. We would like to thank Relja Arandjelović for discussions and for sharing informations regarding $L^3$-Net, and members of the Visual Learning Group at Dartmouth for their feedback.

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
