[Supplementary Material · main_v1.pdf]

# Supplementary material

## 1 Ablation Studies

### 1.1 Performance on downstream tasks as a function of AVTS training epochs

We show the ESC-50 accuracy obtained with a linear SVM trained on our `conv5` audio features for different numbers of AVTS training epochs on the full AudioSet dataset. It can be seen that our procedure yields an improvement of over 30% (80.6% vs 45.2%) compared to features computed from our network randomly initialized (iteration 0).

Figure 1: ESC-50 audio classification accuracy achieved by our `conv5` audio features as a function of number of AVTS training epochs.

### 1.2 Performance on downstream tasks as a function of number of AVTS examples

Table 3 in our paper shows results on audio scene classification for three different AVTS pretraining datasets (Kinetics, AudioSet and SoundNet). In order to understand whether the variations in performance are due to the different content or rather the different sizes of the three datasets, we retrained our AVTS model on a subset of AudioSet having the same size as Kinetics (i.e., 230K videos obtained by randomly sampling 13% of the videos contained in each original AudioSet class). Table 1 below summarizes the results. We include performance also for AudioSet subsets of size 500K and 1M samples. It can be noted that the accuracy achieved using 230K AudioSet samples is similar to that obtained by self-supervised pretraining on Kinetics and that performance increases monotonically with the size of the AudioSet training subset. This suggests that further improvements may come from training on even larger datasets and that our approach is not sensitive to the particular choice of dataset. We have also analyzed how the number of AVTS examples affects action recognition performance (Table 2 in our paper): the UCF101 accuracy is 86.4% when we finetune an MC3 model that was pretrained on 230K AudioSet videos, but it rises to 89% when we pre-train on the full AudioSet of 1.8M samples. This gives an accuracy that is only 1.5% lower (89.0% vs 90.5%) than that obtained by pretraining with full supervision (using action labels) on Kinetics.

## 2 Cross-modal sound localization

Sound localization in video can be achieved by back-propagating audio gradient activations to the video frames using Grad-CAM [1]. Example sound localizations are shown in Fig. 2. Qualitatively, we have found that our model learns to correctly localize the sound but it also often correlates sound with motion. For example, in the Figure we can observe high activations on not only objects that make the sound (such as the accordion in the first row) but also on the moving objects (the hands, and the face in the accordion video).

| Pretraining dataset | # examples (% of dataset) | ESC-50 accuracy (%) | DCASE2014 accuracy (%) |
|---|---|---|---|
| Kinetics | 230K (100%) | 76.7 | 91.2 |
| AudioSet | 230K (13%) | 77.4 | 91.9 |
| AudioSet | 500K (28%) | 78.9 | 92.0 |
| AudioSet | 1M (55%) | 79.5 | 92.6 |
| AudioSet | 1.8M (100%) | 80.6 | 93.1 |

Table 1: Impact of number of AVTS training examples on audio scene classification. Here we vary the AVTS training set size by sampling subsets of AudioSet of different size and by comparing accuracy to that achieved by AVTS audio features trained on the full Kinetics dataset.

Figure 2: Sound localization in video achieved by back-propagating audio gradient activations to the video frames using Grad-CAM [1]. Low activations are filtered for better visualization. Best viewed in color.

# References

[1] Ramprasaath Selvaraju, Michael Cogswell, Abhishek Das, Ramakrishna Vedantam, Devi Parikh, and Dhruv Batra. Grad-cam: Why did you say that? visual explanations from deep networks via gradient-based localization. ICCV, 2017.