[Reviews · NeurIPS 2018]

Reviewer 1



The authors propose self-supervised learning of audio and video features, by means of a curriculum learning setup. In particular, a deep neural network is trained with a contrastive loss function; generating as large feature distances as possible when the video and audio fragment are out of sync, while generating as small distances as possible when they are in sync. The proposed self-supervised learning scheme gives good results for downstream processing, e.g., improving over pure supervised training from scratch. In general, I think this work is highly interesting, since it may significantly reduce the need for manually labelled data sets, and hence may be of significant impact on video & audio processing algorithms. I also think the work is original, although this is not very easy to determine exactly. For instance, the authors mention a very similar paper only in line 258, ref [34]. There may be a clear reason for this: [34] has been published on the Arxiv only in April 2018, so I guess the most likely scenario is that the authors found out about this parallel work very late. I would advise the authors in a revised version though, to mention this work higher up in the paper, perhaps moving the whole related work section there. Still, as the authors claim, the difference with [34] seems to lie in harder examples that are provided in a curriculum learning scheme and hence better results. The authors make an important point that self-supervised learning in pre-training may lead to improved results (further approaching fully supervised learning results), when a bigger training set is given. Now this seems to be supported by results in Table 3, where a higher number of auxiliary samples leads to a higher performance. Still, in this case, the number of samples is not the only difference, it is also the type of dataset. It would be interesting to also see results for different numbers of examples from the same set. There is a result, which I believe to be interesting, but which I think strictly speaking requires more evidence. In Table 3 the results of the "AVTS features" are mentioned. However, it would be good to compare it with another approach that directly applies an SVM to a hidden layer. How good does the SVM work if the activations are taken of a randomly initialized network? This could quantify how much the self-supervised learning improves the features for this task. In summary, a very nice paper that in my eyes should be accepted. Some additional smaller remarks can be found below that aim to further improve the clarity. ----- 1. Line 138 : "but the objective was also difficult to optimize." Can you specify what you mean with this? Did learning not converge in that case? 2. Line 141: you say that it led to better results in every quantitative aspec, also the "feature quality". How was this quality determined, i.e., to what results are you then referring? The SVM ones? 3. The main thing not clear to me for the neural network, is how the initial convolution is done with the N stacked video frames. Is the 3D convolution over all stacked frames together? Then again, the filter is described with 3x7x7, which would suggest that it only goes 3 deep. It is probably described in [19], but I think it would be good to clearly explain it also in this article. 4. Some of the results discussed in 202-207 do not seem to be in table 1. 5. Figure 3 is interesting, although it seems to mostly just take optic flow into account. Still, it does not seem to be referred to, nor discussed in the text. 6. Autoencoders in my eyes also use self-supervised learning, as the supervised targets are self-provided. 7. "in synch" -> "in sync" 8. "One may be argued" -> "One may argue"

Reviewer 2



This paper proposes a self-supervised audio-visual synchronization task for learning high-level audio and video features. The proposed method builds upon [22] by including the motion information (input is a set of frames instead of a single frame as in [22]) through 3D convolutional networks. They present state of the art performance for environmental sound recognition and competitive results for action recognition although the method mainly relies on self-supervision (as opposed to the SOTA I3D-RGB[18]). In overall the idea is straight-forward (using a set of frames instead of a single frame as in [22]), however figuring out how to make it work requires overcoming a set of challenges that paper clearly elaborates. In summary it is a simple idea but executed elegantly. Positives: + Making use of video instead of a single frame for temporal alignment prediction. + Showing that hard-negatives can significantly improve the quality of features when learned in a curriculum based method. + Near state of the art results for environmental audio classification + State of the art results for self-supervised action recognition (UCF101, HMDB51) + The paper has a clear presentation Negatives: - Not too much novelty in the idea, but good execution - Over the limit (9 pages instead of 8) Experiments + It would be helpful to include cross-entropy only learning method in at least one of the tasks. + If possible, it would be good the see the performance of the AVTS objective trained on the I3D [18] model. This result can be included to table 2. References: Would be good to include another concurrent work, though in a different audio-visual setting: https://arxiv.org/abs/1805.11592 typos: line 28: argued -> argue line 56: that that -> that line 76: detail -> detailed Figure 3 is not referenced in text Post-rebuttal update: I think authors submitted a decent rebuttal touching on many concerns of the reviewers. In overall, I think the method is a straight-forward extension of existing work [22] but the execution of the idea is good and it also has comprehensive experimental comparisons with great outcomes. I will keep my score positive.

Reviewer 3



This paper presents an approach for automatically extracting video and audio features by training a self-supervised audio-video synchronizing DNN. The authors did an extensive experimental study, the results on synchronizing and multi-modal action recognition showed the effectiveness of the unsupervised learned features. The language of this paper is articulate, the proposed method is simple and effective, and the references give a broad coverage of the relevant literature. But the paper in its present form has a variety of problems. Firstly, the learning approach should not be called "co-training"--which is an unsupervised multi-view learning approach that first learns a separate classifier for each view using labeled examples, and then the most confident predictions of each classifier on the unlabelled data are then used to iteratively construct additional labeled training data. Hence the proposed A-V synchronizing approach is not a co-training algorithm. Secondly, the idea of shuffling original data to automatically construct supervised data sets and feeding them to a supervised learning system is a natural idea, which has been attempted before, for example: Ishan Misra, C. Lawrence Zitnick, Martial Hebert. Shuffle and Learn: Unsupervised Learning using Temporal Order Verification. ECCV 2016. Andrew Owens, Alexei A. Efros. Audio-Visual Scene Analysis with Self-Supervised Multisensory Features. arXiv:1804.03641, 2018. I was also confused by the structure of the paper. Although section 3 is about experiments, section 2.6 already talks about the implementation and parameters of the experiments, line 143 (which is also in section 2) even introduces the experimental results. Minor questions: line 84: "in synch" -> "in sync"? line 250: what are the other self-supervised approaches, are they implemented like the proposed AVTS approach (i.e., construct negatives by wrongly aligning the A-V pairs)? Table 1: Why did the authors choose 75% easy + 25% hard in the curriculum experiments? How to choose the proportion of the hard negatives?